# Comparative Analysis of Construction-Related Air Pollution in Indoor and Outdoor Environments

Rishikesh Bose, Om Kathalkar and Shreyash Gujar
International Institute of Information Technology, Hyderabad

**Abstract**

The issue at hand revolves around assessing the impact of air pollution emanating from a construction site located approximately 300 meters away from residential buildings, particularly in terms of its effect on human health. This problem is of paramount significance due to the potential health implications for residents living in proximity to the construction site. It also holds relevance for urban planners, environmentalists, and policymakers striving to create sustainable and healthy living environments. Our proposed solution involves deploying air quality monitoring devices both indoors and outdoors in adjacent blocks near the construction area, as well as in a non-constructional area for comparative analysis. We aim to gauge various air quality parameters, including PM2.5, PM10, CO2 levels, temperature, and humidity, and employ statistical tools like correlation coefficients and box plots for comprehensive assessment. The preliminary findings indicate that the construction site not only impacts the closest block but also extends its influence to a block further away, highlighting the far-reaching consequences of such activities. These results underscore the urgency of addressing residents' concerns regarding compromised air quality and associated health risks. In essence, our solution provides valuable insights into the extent of air pollution caused by construction operations, which can inform policies and practices for healthier urban development.

## 1   Introduction

1. What is the problem you are trying to solve?
   In recent times, people have become increasingly concerned about the pollution caused by construction work in cities, especially in their own localities. These pollution sources can be divided into two main types: pollution that comes from fixed sources like chimneys, factories, and construction sites, and pollution from mobile sources, primarily vehicles. Construction sites, falling under the category of fixed sources, are one of the most common and significant contributors to this pollution. Consequently, this pollution not only poses a threat to the health of residents but also exerts a range of adverse effects on the environment, impacting the nearby locality.

2. Why is this problem important, and to whom?
   The problem of construction-related air pollution is vital because it directly impacts the well-being of people living in urban areas. Poor air quality from construction activities can lead to health issues and reduced quality of life for residents. This issue also concerns urban planners and policymakers who strive to create healthier cities. It will push the construction industry to adopt safer and more environmentally friendly practices. Researchers benefit from a deeper understanding of this problem, while regulatory bodies can use the findings to protect public health through better regulations.

3. What is your solution to this problem?
   This study endeavors to fill a research gap by thoroughly investigating the impact of construction-related pollution on residential buildings situated in the urban environment of Hyderabad, India. To accomplish this, the research utilizes seven IoT-based air pollution nodes, consisting of four outdoor and three indoor sensors, deployed within a residential apartment complex, collecting data over a period of around 40 days. What sets this research apart is its comprehensive examination of a wider array

Table 1: Timeline of data collection

| Blocks | Floor | 19 Feb. - 2 Apr. 2023 | |
|:---:|:---:|:---|:---|
| | | **Indoor** | **Outdoor** |
| **I** | 5 | No | Yes |
| **J** | 19 | Yes | Yes |
| **M** | 2 | Yes | Yes |
| **N** | 4 | Yes | Yes |
| **B** | 4 | No | No |
| **Note: For B block, data was collected from April 2 - 24** | | | |

of parameters, distinguishing it from prior studies on construction-related pollution in Indian urban settings. This work aims to address the gap in research by comprehensively analyzing construction-related pollution effects on nearby residential buildings in the Indian urban setting of Hyderabad.

4. What are the supporting results you have to show that your solution can solve this problem?
The comprehensive analysis presented in this study offers a range of benefits. Through the utilization of line plots, box plots, and correlation analyses, the data is visually represented, facilitating a clear interpretation of trends in PM2.5, PM10, temperature, humidity, and CO2 levels. This approach provides a solid foundation for establishing statistical relationships between these parameters, shedding light on their interplay. Moreover, by examining both outdoor and indoor air pollution, as well as comparing them in the context of construction-related activities, the study offers a holistic understanding of the environmental impact. The results unequivocally demonstrate that construction activities lead to a significant increase in these key parameters, signifying a notable decline in air quality. This critical insight equips stakeholders, including policymakers, urban planners, and environmentalists, with valuable information to make informed decisions about regulating and mitigating the environmental impact of construction activities. Ultimately, this analysis serves as a pivotal basis for devising strategies and interventions aimed at preserving a healthier environment. Our proposed methodology has been tested on the testing dataset and we report overall 82% accuracy and 81% F1-Score while estimating the AQI for the images collected.

## 2   Goals

- Deploy the air quality sensor nodes both indoors and outdoors in an apartment located near an active construction site to investigate and gain a deeper understanding of the health-related concerns arising from the construction activities.

- The nodes have been strategically deployed in various apartment blocks, each with distinct directions and varying distances from the construction site. Additionally, these nodes are positioned on different floors to assess how factors like distance and floor level impact the observed effects of the construction activities.

- To understand the significant impact of wind ducts on PM2.5 and PM10 values to gain insights into its effects.

- Conduct a comparative analysis of indoor and outdoor pollution levels to establish a correlation within the data.

- Implementing an image-based solution that is more efficient and highly dependable with minimal effort.

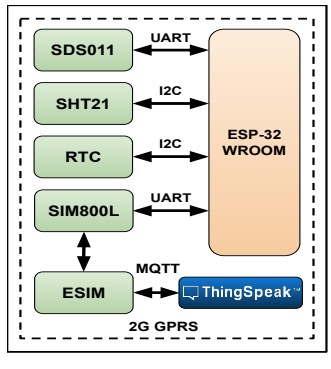 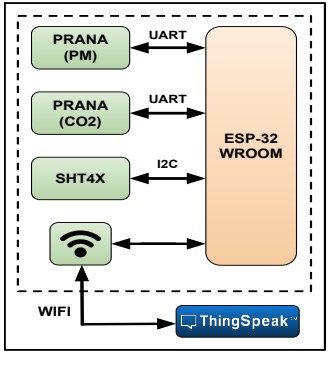 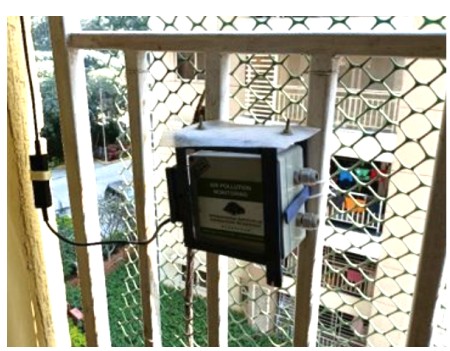

(a) Outdoor node  (b) Indoor node  (c) Actual node view

Figure 1: Outdoor and Indoor node: block diagram and actual view of the deployed node

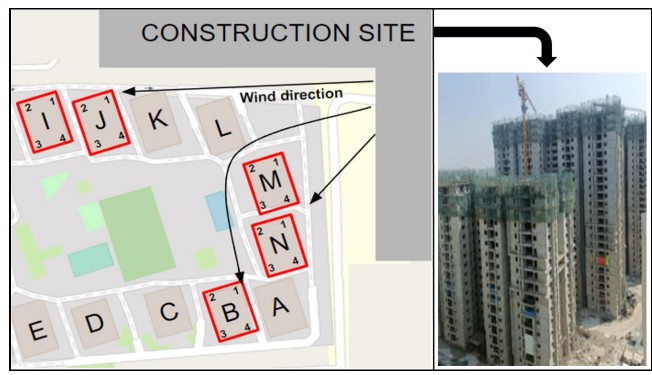 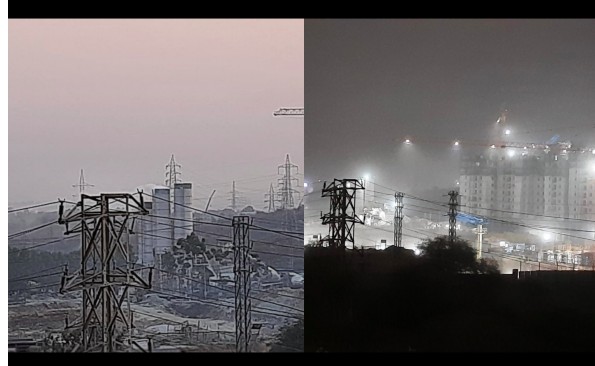

(a) Block view and wind direction from the construction site towards the blocks

(b) Night view of the construction site with the concrete factory (left) which is operated at night time

Figure 2: Construction site and block view during day and night

## 3   System Architecture and Design

### 3.1   Hardware

Figures 1(a) and 1(b) display the schematic representations of the outdoor and indoor nodes, respectively. Figure 1(c) provides an actual visual of the deployed node utilized in the study. The outdoor nodes employed the SDS011 sensor [1] to measure PM2.5 and PM10 levels, while the SHT21 sensor [2] calculated temperature and humidity. Timekeeping was managed by the real-time clock (RTC) module, and the GSM module facilitated data transmission to Thingspeak. On the other hand, the indoor nodes used the Prana PM sensor for PM2.5 and PM10 readings, Prana Air for CO2 monitoring [3], and the Senseair SHT4x [4] sensor for temperature and humidity measurements. Data from the indoor nodes were sent to Thingspeak [5] via a Wi-Fi connection, with a sensing interval of 30 seconds for outdoor nodes and 1 minute for indoor nodes, resulting in approximately 65,000 data points collected over a span of 40 days. Detailed information on the data collection timeline can be found in Table I.

**Construction Site Description:**

Fig. 2 (a) provides a map view of the blocks, where the air pollution nodes were strategically deployed inside an apartment, in Hyderabad, India. The apartment complex comprises several blocks, namely I, J, M, N, and B, situated approximately 300-400 meters away from a construction site. In terms of proximity, the construction site is closest to the N block, followed by M, J, I, and B. The figure also includes some indicators, highlighting the probable wind deviation toward these blocks due to the air flowing from the construction site towards the blocks deviating from the high-rise buildings under construction (The 26-story 14-block construction site has on its north side a lake and on the south a combination of another multi-storied building and an open/green university area. The high-rise buildings surrounding this area leads to wind tunnels).

Also, each block within the image is assigned a number (1,2,3,4 represents residencies), indicating its position relative to the construction site. Notably, residences, numbered 1 and 4 denote that it is facing directly toward the construction site, while residences numbered 2 and 3 show that, the residence is facing opposite to the construction site. This numbering system aids in pro-

## 3.2   Software

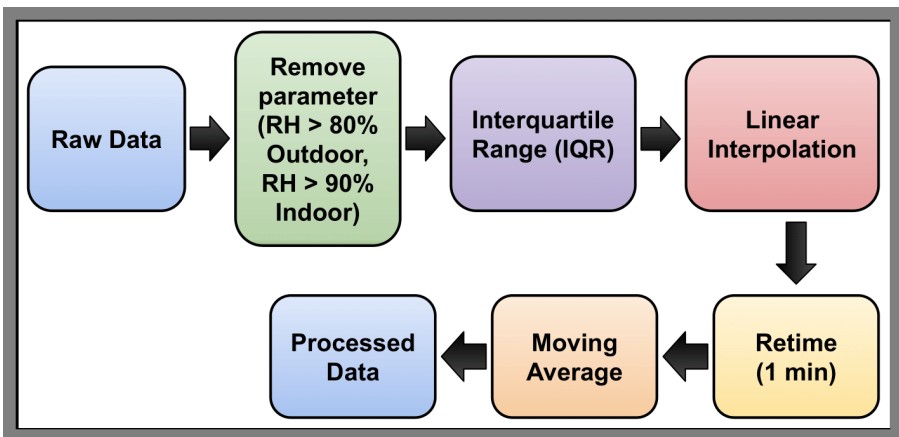

Figure 3: Data pre-processing block

Matlab was used for data pre-processing. Figure 3 outlines the series of steps involved in processing and refining the raw data to ensure its accuracy and reliability. Initially, data points with relative humidity levels exceeding specific thresholds (80 percent for outdoor measurements and 90 percent for indoor measurements) were filtered out. This step aimed to eliminate data influenced by high humidity, which could potentially skew the analysis. Subsequently, outliers were identified and removed using the interquartile range (IQR) method, a statistical technique measuring data spread. This step effectively eliminated abnormal values, preserving data integrity. To address any missing values, linear interpolation was applied, filling gaps in the dataset for a seamless and uninterrupted data series.

Furthermore, the data was standardized to a uniform one-minute interval, ensuring consistency across the dataset. Finally, a moving average approach was implemented to smooth the data, reducing noise and fluctuations. This comprehensive data processing was conducted using Matlab, resulting in a refined dataset suitable for reliable analysis and interpretation. Through these steps, the processed data attained a high level of quality, enabling accurate insights and well-informed decision-making.

## 4   Addressing Challenges

Developing the compact, environmentally resilient device posed multifaceted challenges. It demanded intricate hardware engineering to ensure all components fit without compromising functionality, as well as robust materials and protective measures for durability across diverse environments. Prolonged deployment necessitated careful power management, potentially harnessing solar or battery sources for outdoor use. Establishing stable WiFi connections indoors involved strategic node placement and advanced antenna designs. Balancing security and privacy was paramount, requiring encryption, authentication, and regular audits to thwart breaches. Compliance with data protection laws further complicated deployment, especially across regions with differing legal frameworks. Additionally, remote monitoring, scalability, and user-friendly interfaces were critical considerations, requiring a holistic approach involving hardware, software, and compliance expertise.

## 5   Performance Evaluation and Testing Results

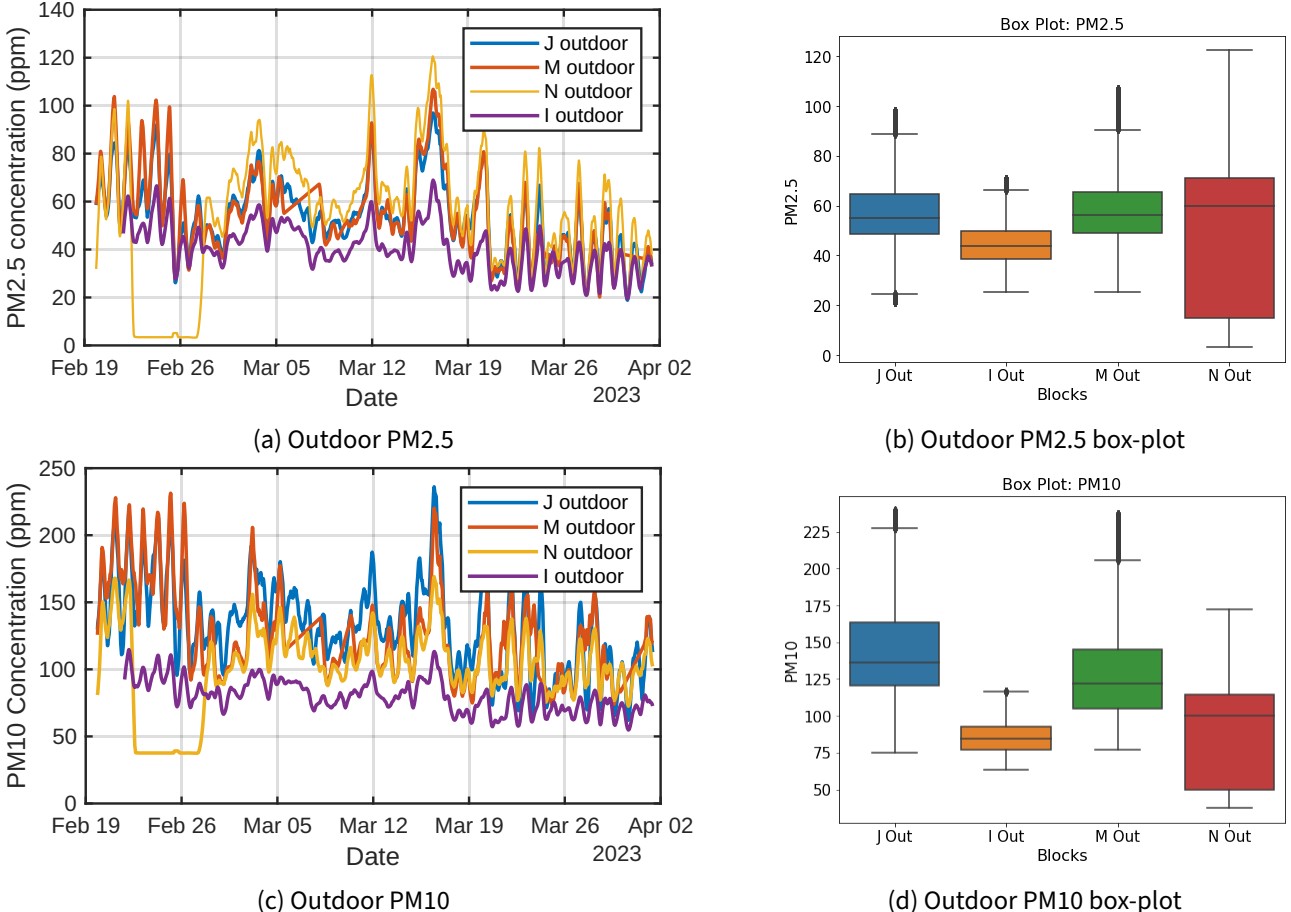

(a) Outdoor PM2.5

(b) Outdoor PM2.5 box-plot

(c) Outdoor PM10

(d) Outdoor PM10 box-plot

Figure 4: The figures (a) and (b) display rising PM2.5 levels exceeding 35 ppm in blocks I, J, M, and N due to construction, with PM10 patterns confirming the impact across these blocks; figures (c) and (d) demonstrate increased outdoor PM10 levels, indicating deteriorating air quality caused by construction, influenced by atmospheric conditions and a wind duct.

## 5.1   Outdoor air pollution analysis

Figures 4(a) and 4(b) illustrate the temporal trends and distribution of PM2.5 pollution levels in blocks I, J, M, and N from February 19th to April 2nd. These visuals reveal a notable increase in pollution levels, surpassing the 35 ppm threshold for both PM2.5 trends and distributions. This rise can be attributed to the ongoing construction activity in the area. Upon analyzing outdoor PM2.5 patterns in blocks J, M, N, and I, it becomes apparent that these regions exhibit similar pollution trends, indicating a uniform impact from the nearby construction work. Additionally, the data indicates that the influence of the construction activity diminishes with increasing distance from the site. Block N, being the closest, experiences the highest pollutant concentration, followed by blocks M, J, and I. This observation is reinforced by the box plots, where the mean PM2.5 values for blocks N, M, J, and I are recorded as 62, 59, 57, and 43, respectively. These results affirm the link between proximity to the construction site and an elevation in outdoor pollution levels.

Figures 4(c) and 4(d) provide compelling evidence that the ongoing construction work significantly impacts outdoor PM10 levels in the nearby regions. The average PM10 values for all the blocks exceeded 80 PPM, indicating a deterioration in air quality due to the construction. Atmospheric conditions, influenced by wind patterns and the presence of a wind duct described in Fig. 2(a), created a scenario where pollutants became concentrated in block J. Consequently, PM10 levels were notably elevated in this area. This relationship is supported by the higher values of PM2.5 observed in block J, which are closer to the PM2.5 values of blocks M and N, regardless of proximity. These findings suggest that both wind patterns and the proximity of the duct to the construction site played a significant role in influencing the pollution levels experienced in block J.

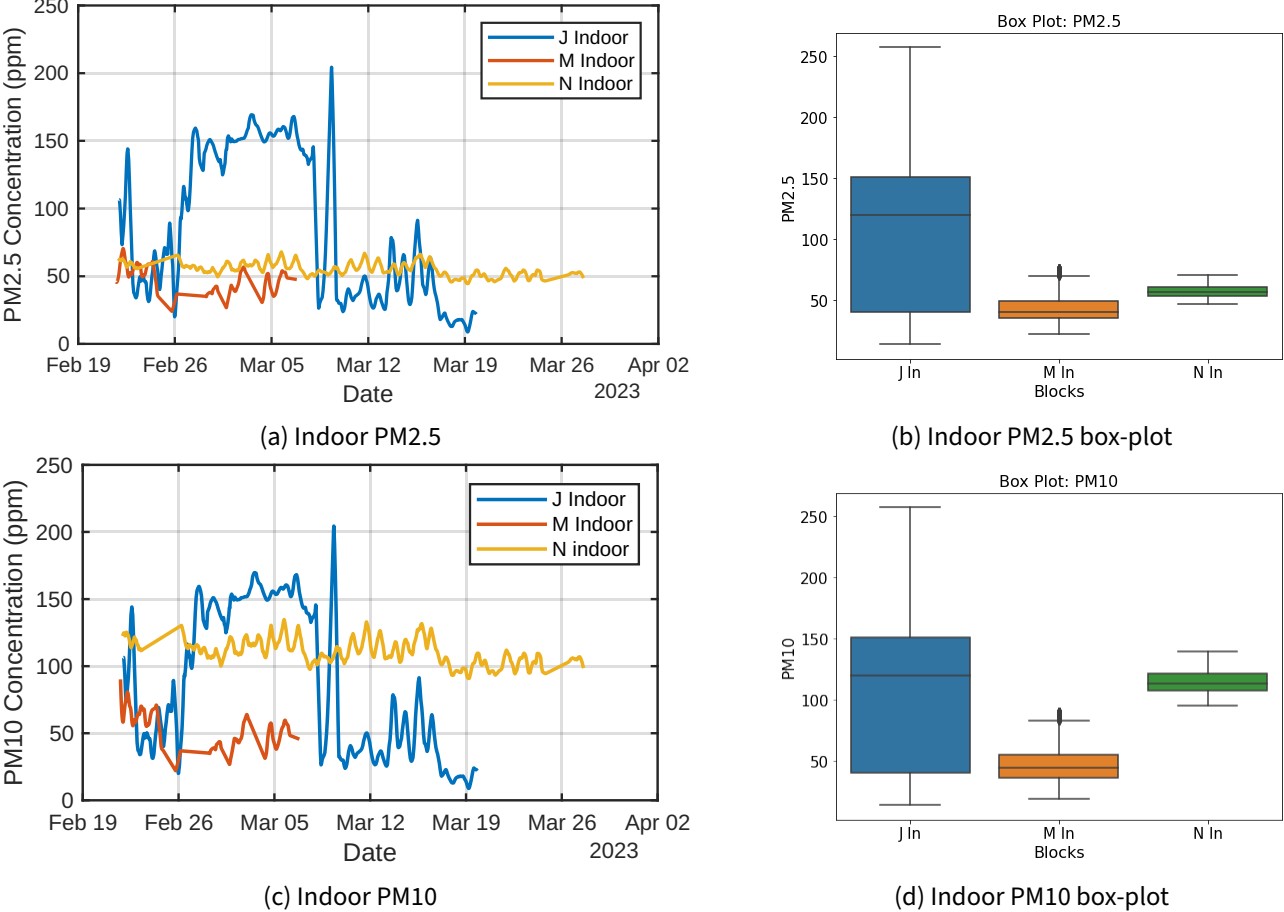

(a) Indoor PM2.5

(b) Indoor PM2.5 box-plot

(c) Indoor PM10

(d) Indoor PM10 box-plot

Figure 5: figures (a) and (c) highlight inadequate ventilation in Block J, particularly for remote workers, while Block N demonstrates the impact of open windows on consistent PM2.5 and PM10 levels from outside sources. Figures (b) and (d) confirm Block J's highest mean values near 125, with a wide range of 50 to 150, suggesting consistent and dynamic indoor activities. Additionally, stable PM10 readings tend to align with similar values for both PM2.5 and PM10, as PM10 includes PM2.5 particles.

## 5.2 Indoor air pollution analysis

Figures 5(a) and 5(c) provide compelling evidence that Block J faces a notable issue of inadequate ventilation, particularly impacting individuals working from their residences. This deficiency in airflow is illustrated by consistently elevated levels of particulate matter (PM2.5 and PM10). Conversely, Figure 5(b) and 5(d) illuminate that in Block N, maintaining open windows results in a steady presence of PM2.5 and PM10, highlighting the influence of external factors on indoor air quality. The data in Figures 5(b) and 5(d) further validate Block J's unique characteristics, with a mean concentration of approximately 125 and a broad range spanning from 50 to 150 for both PM2.5 and PM10. This signifies a sustained and dynamic indoor activity pattern specific to Block J. Additionally, in instances where there are minimal fluctuations in PM10 levels, the sensor readings tend to display similar values for both PM2.5 and PM10, underscoring the encompassing nature of PM10 measurements, which inherently encompass PM2.5 values. These findings collectively emphasize the critical importance of ventilation strategies, especially in densely populated areas like Block J, and highlight the complexities associated with indoor air quality management.

Figure 6 illustrates that among the analyzed blocks, Block J has the poorest ventilation, exacerbated by residents working from home, resulting in higher $CO_2$ levels. Conversely, Block N boasts the most effective ventilation due to open windows, leading to the lowest $CO_2$ emissions. Block M strikes a balance, with moderate ventilation and $CO_2$ concentrations. These variations in ventilation and $CO_2$ levels underscore the disparities in indoor air quality among the individual blocks. Block N exhibits the lowest levels of both pollutants (considering the mean values of PM2.5, PM10, and $CO_2$), while Block M displays somewhat lower $CO_2$ and PM levels compared to Block J. According to this data, the severity of indoor air pollution diminishes

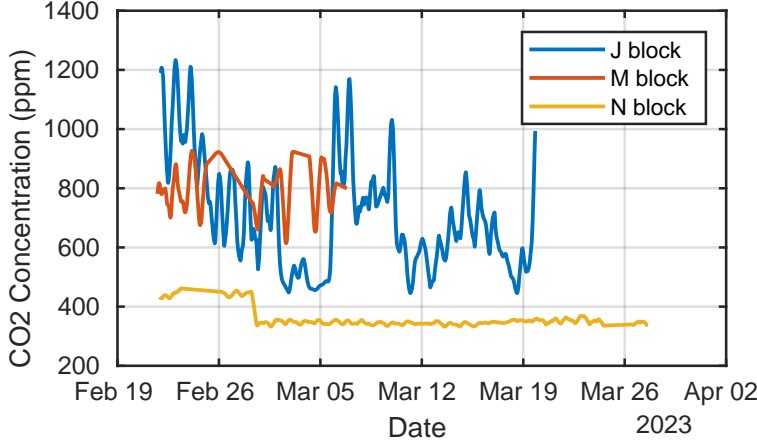

Figure 6: The figure illustrates that Block J has the poorest ventilation and highest CO2 levels due to remote work, while Block N exhibits the best ventilation with the lowest CO2 emissions. Block M represents a balanced situation with moderate ventilation and CO2 concentrations.

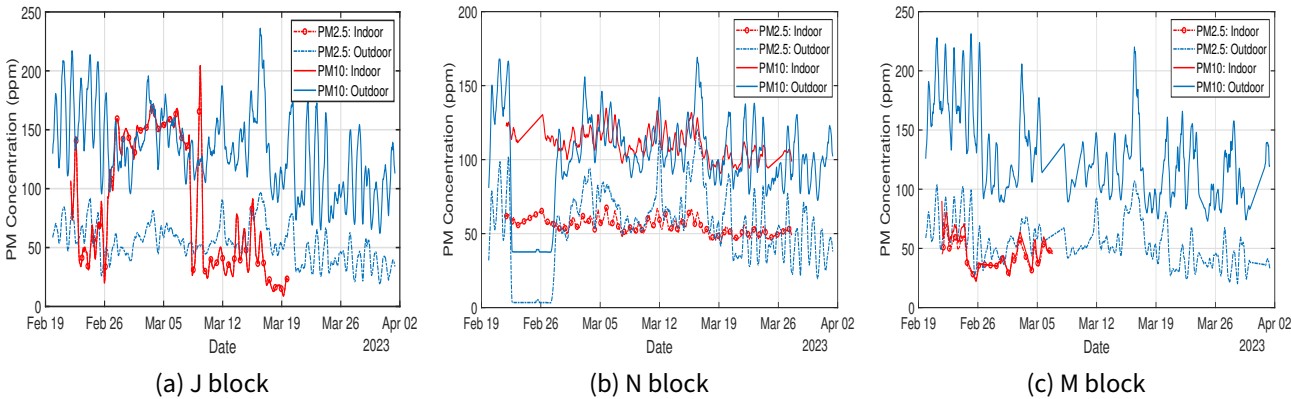

| (a) J block | (b) N block | (c) M block |

Figure 7: Outdoor Vs Indoor (Block wise)

from Block J, being the most affected, to Block M, the next in line, and Block N, with least pollution.

## 5.3 Indoor-Outdoor air pollution comparison

Figure 7(a) illustrates the data trends observed from February 19th to April 2nd, highlighting the consistent patterns between February 26th and March 7th. This indicates that windows were left open for an extended period, leading to a balance between indoor and outdoor PM levels during this specific timeframe.

Figure 7(b) points out that the efficient ventilation in Block N facilitates the entry of outdoor particles into the indoor environment, resulting in a convergence of PM2.5 and PM10 trends between the indoor and outdoor spaces.

Figure 7(c) also demonstrates that the M block, with its moderate ventilation, allows external particles to infiltrate the building.

The correlation coefficients also present valuable insights. This figure 8 unveils novel discoveries that were challenging to articulate solely through time series plots. The correlation between outdoor conditions in block J and indoor conditions in block M, as well as outdoor conditions in block N and indoor conditions in block N, stands at 0.29. This alignment may stem from the proximity of M and N, coupled with the adequate ventilation in N, influencing the correlation within N indoors. Moreover, the correlation between block M and block J outdoor conditions is notable at 0.5. This could be attributed to the prevailing wind direction from the construction site towards the blocks, as depicted in Figure 1(a).

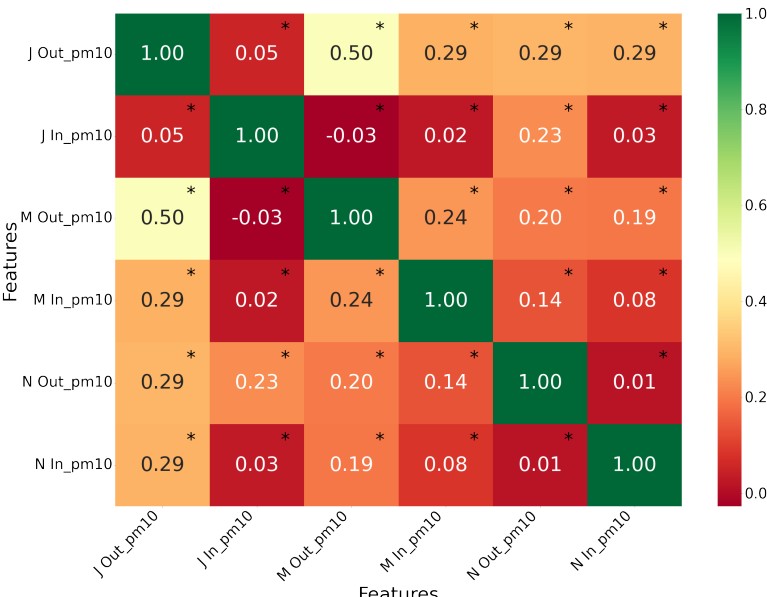

Figure 8: A PM10 correlation heatmap, shows consistent trends with PM2.5. J outdoor correlates with M indoor, N outdoor, and N indoor, due to proximity and better ventilation in N.

## 5.4  Air pollution analysis of the node away from the direct line of site of construction

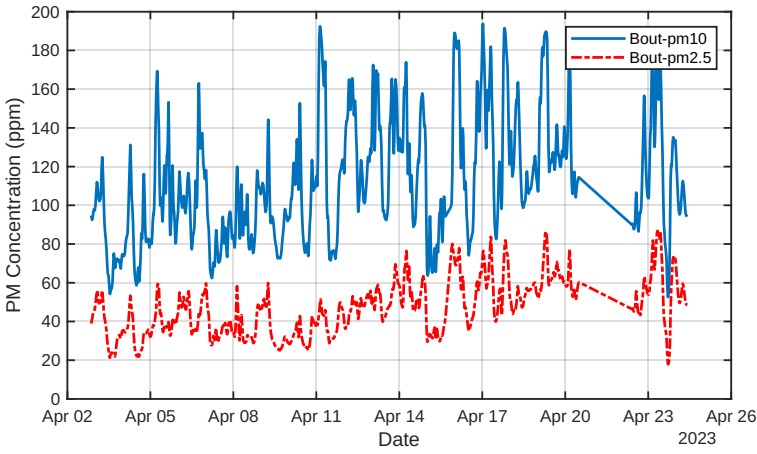

Figure 9: The figure reveals that outdoor PM2.5 in Block B ranges in the 40s, while PM10 levels reach close to 200, attributable to the block's positioning facing the wind duct and receiving airflow that results in elevated PM10 values surpassing the threshold.

Figure 9 offers a detailed representation of the ambient levels of particulate matter (PM2.5 and PM10) outdoors within Block B, which is strategically positioned at a considerable distance from direct exposure to the construction site. Notably, the PM2.5 concentrations are observed to fall within the range of the 40s, indicative of a relatively moderate level of fine particulate matter in the air. In contrast, the PM10 measurements spike significantly, reaching a maximum of 200. This stark disparity in the particulate matter levels can be attributed to Block B's specific orientation, which places it in the path of prevailing wind currents. This leads to an increased flow of air towards Block B, effectively carrying with it higher concentrations of PM10 particles, which tend to be larger and heavier than PM2.5 particles. As a consequence, the PM10 levels surpass the designated threshold, potentially warranting attention and mitigation measures to address the elevated levels of coarse particulate matter in Block B's outdoor environment. This observation highlights the intricate interplay of geographical factors and environmental dynamics in influencing local air quality, emphasizing the need for site-specific assessments in air quality monitoring endeavors.

*Furthermore, we conducted monitoring of Volatile Organic Compounds (VOCs) as an indoor parameter. However, the VOC data exhibited notable variability, presenting a challenge in extracting meaningful insights. This erratic nature of the VOC data can be attributed to indoor activities such as cooking, use of candles, and other potential sources, resulting in substantial fluctuations. This variability hampers the ability to discern clear patterns or trends. Therefore, due to the high degree of variability and the absence of distinct patterns or trends, the VOC data cannot be effectively employed for analysis or for drawing conclusive interpretations.*

## 5.5  Image based solution

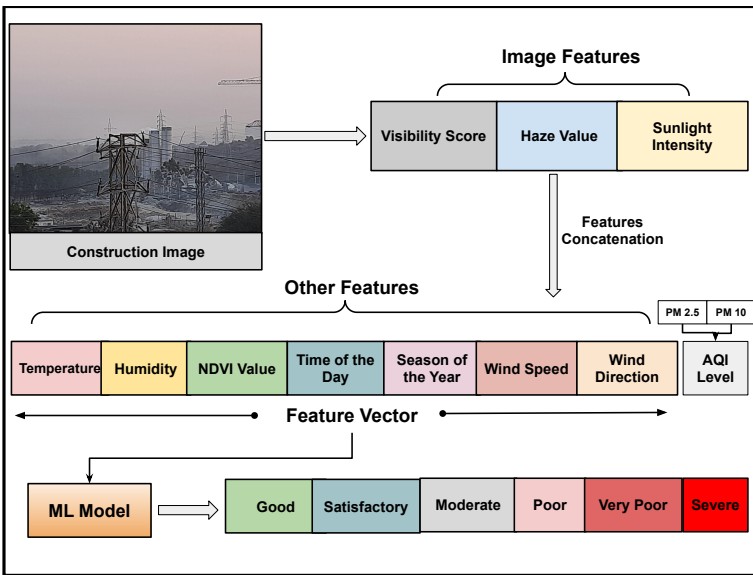

Figure 10: The Pipeline figure for Image-based methodology.

The machine learning (ML) models underwent comprehensive training and validation using the specified dataset, reflecting a meticulous approach to model development. Three distinct models were trained, and their performance is meticulously detailed in Table 2. The Random Forest (RF) model, when applied to the provided dataset, demonstrated a commendable accuracy of 82 percent and an impressive F1-score of 81 percent. This success can be attributed primarily to the robust training process, which ensured that the model was exposed to a diverse range of data points across all categories of Air Quality Index (AQI) levels. This inclusivity spanned from lower AQI values indicating cleaner air, to higher AQI values signifying poorer air quality conditions. The diverse range of data points enabled the RF model to effectively learn and generalize patterns across the entire spectrum of AQI categories, resulting in the high level of accuracy and F1-score observed. This underlines the significance of comprehensive training data that adequately represents the full spectrum of real-world scenarios, ultimately leading to a more robust and reliable machine learning model.

The results obtained for the dataset are presented in Table 2.

Table 2: Performance of various methods on the dataset.

| Method | Dataset | |
|---|---|---|
| | Acc | F1 |
| SVM | 0.77 | 0.76 |
| MLP | 0.79 | 0.78 |
| RF | **0.82** | **0.81** |

## 6    Concluding Remarks and Avenues for Future Work

Construction activities have impacted multiple blocks, with Block N experiencing the highest concentration of outdoor PM10. This emphasizes the differing degrees of influence from nearby construction, notably Block J's inadequate ventilation and Block N's effectiveness in maintaining open windows to mitigate pollution. The duration of window opening in Block J corresponds with the balance between indoor and outdoor PM levels. Disparities in ventilation and CO2 levels across various blocks signify variations in indoor air quality, with Block J exhibiting the highest pollution levels. The correlation analyses and spatial proximity offer valuable insights into how air quality trends interrelate among the outdoor blocks. Stronger associations are observed between specific blocks, regardless of their closeness to the construction site. Notably, Block N experiences a lesser impact compared to Block B, highlighting the role of wind ducts and particle dispersion in shaping air quality. These findings underscore the importance of implementing suitable ventilation and mitigation measures in construction zones to mitigate adverse effects on air quality and public health.

None of the authors oppose construction activity or the economic growth of the area, state, or country. The data collection served as a means to quantify the discomfort experienced by residents in nearby areas due to construction dust. The authors also acknowledge that this discomfort is temporary but stress that children and adults with respiratory conditions like asthma are particularly vulnerable. The authors advocate for policies or technologies that can assist construction companies in reducing dust dispersion near their sites. Furthermore, the pollution resulting from the chemical composition of construction materials also affects personnel involved in these activities, including site engineers, supervisors, and workers.

## 7    Availability

1. Demonstration video

2. Source code

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
