# OpenReview forum: "Comparative Analysis of Construction-Related Air Pollution in Indoor and Outdoor Environments"
_helsinki.fi/ESPC/2023/Competition — ESPC 2023 ShortPresentation_

### Official Review · Reviewer_pRLj · 2023-11-14

**Rating:** 3
**Confidence:** 3

**Summary:**

The article presents the design, installation, and analysis of the air quality monitoring system in the construction area. In particular, seven sensor boxes were installed (indoors and outdoors on different buildings) measuring various air quality parameters. Explorative data analysis is presented. In addition, authors provide an image-based solution.

**Strengths:**

1. Real-world deployment of air quality monitoring in the construction area.
2. Sensor nodes are deployed on different buildings to see the effect of the distance and other environmental factors (like air flow) from the construction area.
3. Continuous data collection for 40 days.

**Weaknesses:**

Overall, authors provide interesting work. There are few issues though,
1. There are many strong claims in the report without proper exploration, analysis, or related work review (some examples: "more efficient", "comprehensively analyzing", "to fill a research gap").
2. Technical notes, how the sensors were calibrated?
3. It is not clear how the wind flow of the area of interest was obtained/constructed? References are needed.
4. More details on data preparation would be beneficial, e.g. how much missing data there were? why linear interpolation was used?
5. In data analysis, it would be good to provide the normative references for accepted air quality references.
6. Indoor air pollution, how do authors know about "maintaining open windows"? is it guess based on the sensor readings?
7. Indoor air pollution, appears that a great portion of indoor data from M block was missing, therefore, authors cannot really make conclusions for that one.
8. I'm not sure what the authors try to accomplish by correlation coefficient comparison.
9. Image based solution requires much greater explanation. Current report gives quite general overview. What is the purpose, how the features were selected from the images. Which models were tried, how the ML training was conducted, how the model performance was evaluated.

Overall, I think that the work is interesting and could be presented at the event. However, I would suggest improving the final report.

---

### Official Review · Reviewer_nAUi · 2023-11-16

**Rating:** 3
**Confidence:** 3

**Summary:**

The authors present a system to monitor the indoor and outdoor air quality and quantify the impact of site location, wind, and the state of the windows (open/close) in buildings close to the construction site. They also performed a study to quantify the effectiveness of using low cost cameras to predict the air quality.

**Strengths:**

- The problem is introduced nicely, and addressing this problem is increasingly becoming important.
- The hardware components used, and the schematic diagrams, are presented in sufficient details.
- The results highlight the impact ventilation systems have on the indoor air quality next to construction sites.

**Weaknesses:**

Answers to the following questions might be helpful to strengthen the report and the work done.
- How was the wind direction measured, and when was the measurement taken (for figure 2)? What were the changes in the wind direction during the days in which the measurements were taken?
- What were the reasons for the outliers?
- Are there any specific reasons why the date ranges of block B are different from the other blocks?
- How did you quantify the accuracy of the values being reported? Was there any ground truth which was used to quantify the accuracy of the measured values?
- How did the authors monitor the state of the windows at each of the sites?
- How was the haze value, visibility score, and sunlight intensity computed/measured?

Minor nitpicks:
- When making a claim of "comprehensive examination of a wider array of parameters", it is essential to specify the state-of-the-art against which the claim is being made.
- Specifying the camera settings such as exposure time, aperture, etc., would be helpful for the readers to get insights on the quality of the image that is being analyzed.
- Please ensure that the Y-axis ranges are the same in Figure 7. (https://www.callingbullshit.org/tools/tools_misleading_axes.html)

---

### Official Review · Reviewer_yxwJ · 2023-11-18

**Rating:** 2
**Confidence:** 4

**Summary:**

This project presents air pollution data analytics for indoor residential buildings and outdoor environments that is a construction site. The project deploys a few sensors and collects PM2.5, PM10, CO2, temperature, and humidity variables; and images taken by a Raspberry Pi. The project carries out normal data analytics work and leaves the image analysis for future work. In its current form, it is a nicely written report that deploys sensors, collects data, and carries out simple data analysis.

**Strengths:**

The report is nicely written and presents data analytics for indoor and outdoor.

**Weaknesses:**

The report in its current form is only at the data analytics level. I was expecting to see the use of image data and the related results in the report. In the matrix plot that is for correlation analysis, the report does not say what correlation method (e.g., Pearson correlation) it uses and why it uses the specific correlation method.
The report and the video do not emphasize and do not provide further details about the hardware and software used in the project. It seems the novelty part will be the use of camera images that are taken by RPi, which has been postponed as future work.

---

### Official Review · Reviewer_u1w5 · 2023-11-18

**Rating:** 3
**Confidence:** 4

**Summary:**

In this work, authors have picked up an interesting problem of air pollution caused due to construction of buildings. They consider the impact of construction with respect to pollution from both indoor and outdoor perspective. This problem is prevalent in busy and congested cities.
Authors have developed seven IoT nodes for monitoring this kind of pollution -- four of which are deployed outdoors and three indoors.
The outdoor nodes use SDS011 sensor for measuring PM2.5 and PM10 levels, SHT21 sensor for measuring temperature and humidity, GSM module for uploading data to IoT cloud -- Thingspeak. The indoor nodes use the Prana PM sensor for measuring PM2.5 and PM10 readings, Prana Air sensor for measuring CO2, and the Senseair SHT4x sensor for measuring temperature and humidity. Indoor nodes send data to IoT cloud -- Thingspeak over WiFi.
They collected data for 40 days and present its analysis results for the following metrics -- PM2.5, PM10, temperature, humidity, and CO2 levels.
They observe that these construction activities significantly reduce air quality.

**Strengths:**

*They have targeted a pressing problem of busy cities
*Real-world data collected and analyzed
*Detailed statistical analysis of data presented
*Interesting insights shown on the data collected
*Presented an image based solution as well monitoring the pollution based on image features such as, visibility score, haze, and sunlight intensity.

**Weaknesses:**

--Missing explanation on why separate indoor and outdoor nodes are needed.
--They have given an image based solution as well but there is no discussion on how these images are collected. The node prototype does not have any camera installed as well.
--Accuracy of models should be improved ($\approx 80$\%, currently)
--A discussion on how to address the problem would have helped

---

### Official Review · Reviewer_txYP · 2023-11-20

**Rating:** 3
**Confidence:** 4

**Summary:**

The  project assesses the impact of air pollution from a construction site located approximately 300 meters away from residential buildings.  The team deployed air quality monitoring devices indoors and outdoors in adjacent blocks near the construction area, and in a non-constructional area for comparative analysis. The findings indicate that the construction site impacted the closest block and extended its influence to a block further away, This a good topic and the findings are interesting. The report could have been better structured.

**Strengths:**

It is a good to read about the effects of construction site on exposure to air pollution living in apartment blocks (both outdoor and indoor) at different distances to the source. It is good to see measurements from India experiencing very high PM2.5 concentrations. The report is acceptance: introduction is set out in a series of questions and goals are set out well. The system designs section is informative. There are informative graphs in the outdoor and indoor air pollution analysis section. The general findings are interesting but not well structured.    The video is well balanced with different team members describing their work., Thank you for sharing the code.

**Weaknesses:**

It is good report but the sections are mixed up. There are no references in the Introduction to generated a research problem. The method  sections could have been structured as an experimental design. A better description of the blocks and distance to the construction site in form of table or detailed diagram would of helped with the results and analysis. The challenges section should be part of the introduction. The image processing sections seems to be inserted as an after thought.